# CHAMELEON: INCREASING LABEL-ONLY MEMBERSHIP LEAKAGE WITH ADAPTIVE POISONING

**Harsh Chaudhari, Giorgio Severi, Alina Oprea, Jonathan Ullman**
Khoury College of Computer Sciences
Northeastern University
Boston, MA 02115, USA
`{chaudhari.ha, severi.g, a.oprea, j.ullman}@northeastern.edu`

## ABSTRACT

The integration of machine learning (ML) in numerous critical applications introduces a range of privacy concerns for individuals who provide their datasets for model training. One such privacy risk is Membership Inference (MI), in which an attacker seeks to determine whether a particular data sample was included in the training dataset of a model. Current state-of-the-art MI attacks capitalize on access to the model's predicted confidence scores to successfully perform membership inference, and employ data poisoning to further enhance their effectiveness. In this work, we focus on the less explored and more realistic *label-only* setting, where the model provides only the predicted label on a queried sample. We show that existing label-only MI attacks are ineffective at inferring membership in the low False Positive Rate (FPR) regime. To address this challenge, we propose a new attack Chameleon that leverages a novel adaptive data poisoning strategy and an efficient query selection method to achieve significantly more accurate membership inference than existing label-only attacks, especially at low FPRs.

## 1 INTRODUCTION

The use of machine learning for training on confidential or sensitive data, such as medical records (Stanfill et al., 2010), financial documents (Ngai et al., 2011), and conversations (Carlini et al., 2021), introduces a range of privacy violations. By interacting with a trained ML model, an attacker might reconstruct data from the training set (Haim et al., 2022; Balle et al., 2022), perform membership inference (Shokri et al., 2017; Yeom et al., 2018; Carlini et al., 2022), or learn sensitive attributes from training data (Fredrikson et al., 2015; Mehnaz et al., 2022). Membership inference (MI) attacks (Shokri et al., 2017), originally introduced under the name of tracing attacks (Homer et al., 2008), enable an attacker to determine whether or not a data sample was included in the training set of an ML model. While these attacks are less severe than training data reconstruction, they might still constitute a serious privacy violation. Consider a mental health clinic that uses an ML model to predict patient treatment responses based on medical histories. An attacker with accesses to a certain individual's medical history can learn if the individual has a mental health condition by performing a successful MI attack.

We can categorize MI attacks into two groups: confidence-based attacks in which the attacker gets access to the target ML model's predicted confidences, and label-only attacks, in which the attacker obtains only the predicted label for queried samples. Recent literature has primarily focused on confidence-based attacks Carlini et al. (2022); Bertran et al. (2023) that maximize the attacker's success at low False-Positive Rates (FPRs). Additionally, Tramèr et al. (2022) and Chen et al. (2022) showed that introducing data poisoning during training significantly improves the MI performance at low FPRs in the confidence-based scenario.

Nevertheless, in many real-world scenarios, organizations training ML models respond only with hard labels to customer queries. For example, financial institutions might solely indicate whether a customer has been granted a home loan or credit card approval. In such cases, launching an MI attack gets considerably more challenging as the attacker looses access to prediction confidences and cannot leverage state-of-the-art attacks such as Carlini et al. (2022), Wen et al. (2023), Bertran et al. (2023).

Furthermore, it remains unclear whether existing label-only MI attacks, such as Choquette-Choo et al. (2021) and Li & Zhang (2021), are effective in the low FPR regime and if data poisoning techniques can be used to amplify the membership leakage in this specific realistic scenario.

In this paper, we first show that existing label-only MI attacks (Yeom et al., 2018; Choquette-Choo et al., 2021; Li & Zhang, 2021) struggle to achieve high True Positive Rate (TPR) in the low FPR regime. We then demonstrate that integrating state-of-the-art data poisoning technique (Tramèr et al., 2022) into these label-only MI attacks further degrades their performance, resulting in even lower TPR values at the same FPR. We investigate the source of this failure and propose a new label-only MI attack Chameleon that leverages a novel *adaptive* poisoning strategy to enhance membership inference leakage in the label-only setting. Our attack also uses an *efficient* querying strategy, which requires only 64 queries to the target model to succeed in the distinguishing test, unlike prior works (Choquette-Choo et al., 2021; Li & Zhang, 2021) that use on the order of a few thousand queries. Extensive experimentation across multiple datasets shows that our Chameleon attack consistently outperforms previous label-only MI attacks, with improvements in TPR at 1% FPR ranging up to $17.5\times$. Finally, we also provide a theoretical analysis that sheds light on how data poisoning amplifies membership leakage in label-only scenarios. To the best of our knowledge, this work represents the first analysis on understanding the impact of poisoning on MI attacks.

## 2 BACKGROUND AND THREAT MODEL

We provide background on membership inference, describe our label-only threat model with poisoning, and analyze existing approaches to motivate our new attack.

**Related Work.** *Membership Inference* attacks can be characterized into different types based on the level of adversarial knowledge required for the attack. Full-knowledge (or white-box) attacks (Nasr et al., 2018; Leino & Fredrikson, 2020) assume the adversary has access to the internal weights of the model, and therefore the activation values of each layer. In black-box settings the adversary can only query the ML model, for instance through an API, which may return either confidence scores or hard labels. The confidence setting has been studied most, with works like Shokri et al. (2017); Carlini et al. (2022); Ye et al. (2022) training multiple shadow models —local surrogate models— and modeling the loss (or logit) distributions for members and non-members.

The *label-only* MI setting, investigated by Yeom et al. (2018); Choquette-Choo et al. (2021); Li & Zhang (2021), considers a more realistic threat model that returns only the predicted label on a queried sample. Designing MI attacks under this threat model is more challenging, as the attack cannot rely on separating the model's confidence on members and non-members. Existing label-only MI attacks are based on analyzing the effects of perturbations on the original point on the model's decision. With our work we aim to improve the understanding of MI in the label-only setting, especially in light of recent trends in MI literature.

Current MI research, in fact, is shifting the attention towards attacks that achieve high True Positive Rates (TPR) in low False Positive Rates (FPR) regimes (Carlini et al., 2022; Ye et al., 2022; Liu et al., 2022; Wen et al., 2023; Bertran et al., 2023). These recent papers argue that if an attack can manage to *reliably* breach the privacy of even a small number of, potentially vulnerable, users, it is still extremely relevant, despite resulting in potentially lower average-case success rates. A second influential research thread exposed the effect that training data poisoning has on amplifying privacy risks. This threat model is particularly relevant when the training data is crowd-sourced, or obtained through automated crawling (common for large datasets), as well as in collaborative learning settings. Tramèr et al. (2022) and Chen et al. (2022) showed that data poisoning amplifies MI privacy leakage and increases the TPR values at low FPRs. Both LiRA (Carlini et al., 2022) and Truth Serum (Tramèr et al., 2022) use a large number of shadow models (typically 128) to learn the distribution of model confidences, but these methods do not directly apply to label-only membership inference, a much more challenging setting. A related line of research (Mahloujifar et al., 2022; Chaudhari et al., 2023) showcased how data poisoning could be utilized to infer statistical information about the properties of the training set, called property inference attacks.

**Threat Model.** We follow the threat model of Tramèr et al. (2022) used for membership inference with data poisoning, with adjustments to account for the more realistic label-only setting. The attacker has black-box query access —the ability to send samples and obtain the corresponding outputs— to a

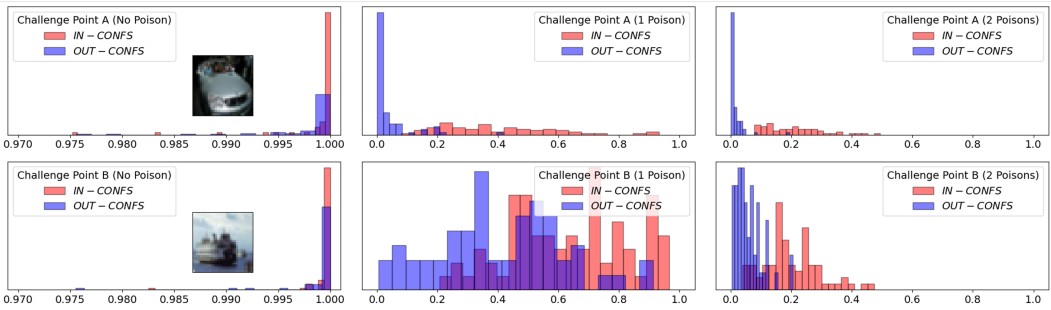

Figure 1: Distribution of confidence scores for two challenge points (a Car and a Ship), highlighting the impact of poisoning on two different points of the CIFAR-10 dataset. Each row shows the shift in model confidences (wrt. true label) for IN and OUT models with introduction of poisoned samples.

trained machine learning model $M_t$, also called target model, that returns only the predicted label on an input query. The attacker's objective is to determine whether a particular *challenge point* was part of $M_t$'s training set or not. Similarly to Tramèr et al. (2022) the attacker $\mathcal{A}$ has the capability to inject additional poisoned data $D_p$ into the training data $D_{tr}$ sampled from a data distribution $\mathcal{D}$. The attacker can only inject $D_p$ once before the training process begins, and the adversary does not participate further in the training process after injecting the poisoned samples. After training completes, the adversary can only interact with the final trained model to obtain predicted labels on selected queried samples. Following the MI literature, $\mathcal{A}$ can also train local shadow models with data from the same distribution as the target model's training set. Shadow models training sets may or may not include the challenge points. We will call IN models those trained with the challenge point included in the training set, and OUT models those trained without the challenge point.

**Analyzing Existing Approaches.** Existing label-only MI attacks (Choquette-Choo et al., 2021; Li & Zhang, 2021) propose a decision boundary technique that exploits the existence of adversarial examples to create their distinguishing test. These approaches typically require a large number of queries to the target model to estimate a sample's distance to the model decision boundary. However, these attacks achieve low TPR (e.g., 1.1%) at 1% FPR , when tested on the CIFAR-10 dataset. In contrast, the LiRA confidence-based attack by Carlini et al. (2022) achieves a TPR of 16.2% at 1% FPR on the same dataset. Truth Serum (Tramèr et al., 2022) evaluates LiRA with a data poisoning strategy based on label flipping, which significantly increases TPR to 91.4% at 1% FPR once 8 poisoned samples are inserted per challenge point.

A natural first strategy for label-only MI with poisoning is to incorporate the Truth Serum data poisoning method to the existing label-only MI attack (Choquette-Choo et al., 2021) and investigate if the TPR at low FPR can be improved. The Truth Serum poisoning strategy is simply label flipping, where poisoned samples have identical features to the challenge point, but a different label. Surprisingly, the results show a negative outcome, with the TPR decreasing to 0% at 1% FPR after poisoning. This setback compels us to reconsider the role of data poisoning in improving privacy attacks within the label-only MI threat model. We question whether data poisoning can indeed improve label-only MI, and if so, why did our attempt to combine the two approaches fail. In the following section, we provide comprehensive answers to these questions and present a novel poisoning strategy that significantly improves the attack success in the label-only MI setting.

## 3    CHAMELEON ATTACK

We first provide some key insights for our attack, then describe the detailed attack procedure, and include some analysis on leakage under MI.

### 3.1    ATTACK INTUITION

Given the threat model, our goal is to improve the TPR in the *low* FPR regime for label-only MI, while reducing the number of queries to the target model $M_t$. To achieve this two-fold objective, we start by addressing the fundamental question of determining an effective poisoning strategy. This

---

**Algorithm 1** Adaptive Poisoning Strategy

---

**Input:** Challenge point $(x, y)$, poisoned point $(x, y')$ where $y' \neq y$, attacker's dataset $D_{adv}$, poison threshold $t_p$ and maximum poisoned iterations $k_{max}$.

1: Let $k$ denote the number of poisoned replicas.
2: **For** $k = 0, \ldots, k_{max}$ **do:**
3:     Construct poisoned dataset $D_p$ containing $k$ replicas of $(x, y')$.
4:     Train $m$ OUT models $\{\theta_1^{out}, \ldots, \theta_m^{out}\}$ on dataset $D_p \cup D_{adv}$.
5:     Query $x$ on OUT models and obtain confidences $c_1^y, \ldots, c_m^y$ for label $y$, where $0 \leq c_i^y \leq 1$.
6:     Compute mean of the confidences $\mu = \frac{\sum_{i=1}^m c_i^y}{m}$.
7:     **If** $\mu \leq t_p$**:**
8:         **break**
9:     $k = k + 1$
**Output:** Number of poisoned replicas $k$.

---

involves striking the right balance such that the IN models classify the point correctly, and the OUT models misclassify it. Without poisoning, it is likely that both IN and OUT models will classify the point correctly (up to some small training and generalization error). Conversely, if we insert too many poisoned samples with an incorrect label, then both IN and OUT models will misclassify the point to the incorrect label. As the attacker only gets access to the labels of the queried samples from $M_t$, over-poisoning would make it implausible to distinguish whether the model is an IN or OUT model.

The state-of-the-art Truth Serum attack (Tramèr et al., 2022), which requires access to model confidences, employs a static poisoning strategy by adding a fixed set of $k$ poisoned replicas for each challenge point. This poisoning strategy fails for label-only MI as often times both IN and OUT models misclassify the target sample, as discussed in Section 2. Our *crucial* observation is that not all challenge points require the same number of poisoned replicas to create a separation between IN and OUT models. To provide evidence for this insight, we show a visual illustration of model confidences under the same number of poisoned replicas for two challenge points in Figure 1. Therefore, we propose a new strategy that adaptively selects the number of poisoned replicas for each challenge point, with the goal of creating a separation between IN and OUT models. The IN models trained with the challenge point in the training set should classify the point correctly, while the OUT models should misclassify it. Our strategy adaptively adds poisoned replicas until the OUT models consistently misclassify the challenge point at a significantly higher rate than the IN models.

Existing MI attacks with poisoning (Tramèr et al., 2022; Chen et al., 2022) utilize confidence scores obtained from the trained model to build a distinguishing test. As our attacker only obtains the predicted labels, we develop a label-only "proxy" metric for estimating the model's confidence, by leveraging the predictions obtained on "close" neighbors of the challenge point. We introduce the concept of a *membership neighborhood*, which is constructed by selecting the closest neighbors based on the KL divergence computed on model confidences. This systematic selection helps us improve the effectiveness of our attack by strategically incorporating only the relevant neighbor predictions. The final component of the attack is the distinguishing test, in which we compute a score based on the target model $M_t$'s correct predictions on the membership neighborhood set. These scores are used to compute the TPR at fixed FPR values, as well as other metrics of interest such as AUC and MI accuracy. We provide a detailed description for each stage of our attack below.

## 3.2 ATTACK DETAILS

Our Chameleon attack can be described as a three-stage process:

**Adaptive Poisoning.** Given a challenge point $(x, y)$ and access to the underlying training distribution $\mathcal{D}$, the attacker constructs a training dataset $D_{adv} \sim \mathcal{D}$, such that $(x, y) \notin D_{adv}$ and a poisoned replica $(x, y')$, for some label $y' \neq y$. The goal of the attacker is to construct a small enough poisoned set $D_p$ such that a model trained on $D_{adv} \cup D_p$, which *excludes* $(x, y)$, missclassifies the challenge point. The attacker needs to train their own OUT shadow models

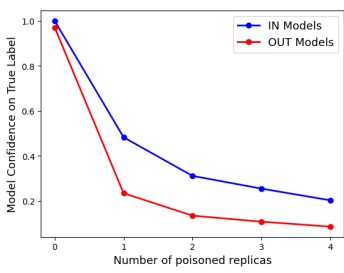

Figure 2: Impact of poisoning on confidences of IN and OUT models (wrt. true label) for a point in CIFAR-10 dataset.

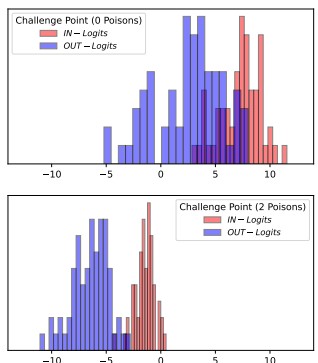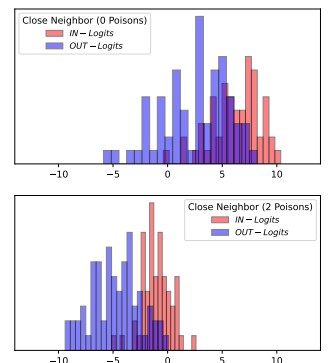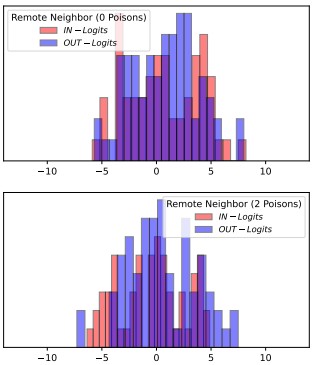

Figure 3: Effect of poisoning on the scaled confidence (logit) distribution of a challenge point and its neighbors. Both IN and OUT distributions of near neighbor, unlike the far-away neighbor, exhibit a behavior similar to the challenge point distribution before and after introduction of poisoning.

(without the challenge point) to determine how many poisoned replicas are enough to mis-classify the challenge point. Using a set of $m$ OUT models instead of a single one increases the chance that any other OUT model (e.g., the target model $M_t$) has a similar behavior under poisoning. The attacker begins with no poisoned replicas and trains $m$ shadow models on the training set $\mathsf{D}_{\mathsf{adv}}$. The attacker adds a poisoned replica if the average confidence across the OUT models on label $y$ is above a threshold $\mathsf{t}_\mathsf{p}$, and repeats the process until the models' average confidence on label $y$ falls below $\mathsf{t}_\mathsf{p}$ (threshold where mis-classification for the challenge point occurs). The details of our adaptive poisoning strategy are outlined in Algorithm 1, which describes the iterative procedure for constructing the poisoned set.

Note that we *do not* need to separately train any IN models, i.e., models trained on $\mathsf{D}_\mathsf{p} \cup \mathsf{D}_\mathsf{adv} \cup \{(x, y)\}$, to select the number of poisoned replicas for our challenge point. This is due to our observation that, in presence of poisoning, the average confidence for the true label $y$ tends to be higher on the IN models when compared to the OUT models. Figure 2 illustrates an instance of this phenomenon, where the average confidence on the OUT models decreases at a faster rate than the confidence on the IN models with the addition of more poisoned replicas. As a result, the confidence mean computed on the OUT models (line 6 in Algorithm 1) will always cross the poisoning threshold $\mathsf{t}_\mathsf{p}$ first, leading to misclassification of the challenge point by the OUT models before the IN models. Therefore, we are only required to train OUT models for our adaptive poisoning strategy.

*Multiple Challenge Points:* In practical scenarios, an attacker aims to infer membership across multiple challenge points rather than focusing on a single point. In Appendix C, we propose a strategy that handles a set of challenge points simultaneously while naturally capturing interactions among them during the poisoning phase. Importantly, our strategy only incurs a fixed overhead cost, enabling the attack to scale to any number of challenge points. For resource-limited scenarios, we also propose an optimized variant of our attack requiring just 16 shadow models while outperforming prior label-only attacks. A detailed description and cost analysis is reported in Appendix C and D.

**Membership Neighborhood.** In this stage, the attacker's objective is to create a membership neighborhood set $\mathsf{S}_{\mathsf{nb}}^{(x,y)}$ by selecting close neighboring points to the challenge point. This set is then used to compute a proxy score in order to build a distinguishing test. To construct the neighborhood set, the attacker needs $N$ shadow models such that the challenge point $(x, y)$ appears in the training set of half of them (IN models), and not in the other half (OUT models). Interestingly, the attacker can reuse the OUT models trained from the previous stage and reduce the computational cost of the process. Using these shadow models, the attacker constructs the neighborhood set $\mathsf{S}_{\mathsf{nb}}^{(x,y)}$ for a given challenge point $(x, y)$. A candidate $(x_c, y)$, where $x_c \neq x$, is said to be in set $\mathsf{S}_{\mathsf{nb}}^{(x,y)}$, if the original point and the candidate's model confidences are close in terms of KL divergence for both IN and OUT models, i.e., the following conditions are satisfied:

$$\mathsf{KL}(\ \Phi(x_c)_{\mathsf{IN}}\ ||\ \Phi(x)_{\mathsf{IN}}\ ) \leq \mathsf{t}_{\mathsf{nb}} \text{ and } \mathsf{KL}(\ \Phi(x_c)_{\mathsf{OUT}}\ ||\ \Phi(x)_{\mathsf{OUT}}\ ) \leq \mathsf{t}_{\mathsf{nb}} \tag{1}$$

Here, $\mathsf{KL}()$ calculates the Kullback-Leibler divergence between two distributions. Notations $\Phi(x_c)_{\mathsf{IN}}$ and $\Phi(x_c)_{\mathsf{OUT}}$ represent the distribution of confidences (wrt. label $y$) for candidate $(x_c, y)$ on the IN and OUT models trained with respect to challenge point $(x, y)$.

Note that the models used in this stage do not need to include poisoning into their training data. We observe that the distribution of confidence values for candidates characterized by low KL divergence tend to undergo similar changes as those of the challenge point when poisoning is introduced. We call such candidates *close neighbors*. In Figure 3, we show how the confidence distribution of a close neighbor closely mimics the confidence distribution of the challenge point as we add two poisoned replicas. Additionally, we also demonstrate that the confidence distribution of a *remote neighbor* is hardly affected by the addition of poisoned replicas and does not exhibit a similar shift in its confidence distribution as the challenge point. Therefore, it is enough to train shadow models without poisoning, which reduces the time complexity of the attack.

In practical implementation, we approximate the distributions $\Phi(x_c)_{\text{IN}}$ and $\Phi(x_c)_{\text{OUT}}$ using a scaled version of confidences known as logits. Previous work (Carlini et al., 2022; Tramèr et al., 2022) showed that logits exhibit a Gaussian distribution, and therefore we compute the KL divergence between the challenge point and the candidate confidences using Gaussians. In Section 4.3, we empirically show the importance of selecting close neighbors.

**Distinguishing Test.** The final goal of the attacker is to perform the distinguishing test. Towards this objective, the attacker queries the black-box trained model $M$ using the challenge point and its neighborhood set $\mathsf{S}_{\text{nb}}^{(x,y)}$ consisting of $n$ close neighbors. The attacker obtains a set of predicted labels $\{\hat{y}_1, \ldots, \hat{y}_{n+1}\}$ in return and computes the missclassification score of the trained model $f(x)_y = \frac{\sum_{i=1}^{n+1} \hat{y}_i \neq y}{n+1}$. The score $f(x)_y$ denotes the fraction of neighbors whose predicted labels do not match the ground truth label. This score is then used to predict if the challenge point was a part of the training set or not. Correspondingly, we use the computed misclassification score $f(x)_y$ to calculate various metrics, including TPR@ fixed FPR, AUC, and MI accuracy.

### 3.3 LABEL-ONLY MI ANALYSIS

We now analyze the impact of poisoning on MI leakage in the label-only setting. We construct an *optimal* attack that maximizes the True Positive Rate (TPR) at a fixed FPR of $x\%$, when $k$ poisoned replicas related to a challenge point are introduced in the training set. The formulation of this optimal attack is based on a list of assumptions outlined in Appendix B. The objective of constructing this optimal attack is twofold. First, we aim to examine how increasing the number of poisoned replicas influences the maximum TPR (@$x\%$FPR). Second, we seek to evaluate whether the behavior of Chameleon aligns with (or diverges from) the behavior exhibited by the optimal attack. In Figure 4, we present both attacks at a FPR of 5% on CIFAR-10. The plot depicting the optimal attack shows an initial increase

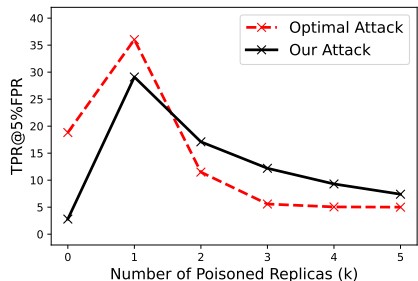

Figure 4: Comparing Optimal and Our attack under poisoning.

in the maximum attainable TPR following the introduction of poisoning. However, as the number of poisoned replicas increases, the TPR decreases, indicating that excessive poisoning in the label-only scenario adversely impacts the attack TPR. Notably, Chameleon exhibits a comparable trend to the optimal attack, showing first an increase and successively a decline in TPR as poisoning increases. This alignment suggests that our attack closely mimics the behavior of the optimal attack and has a similar decline in TPR due to overpoisoning. The details of our underlying assumptions and the optimal attack are given in Appendix B.

## 4 EXPERIMENTS

We show that Chameleon significantly improves upon prior label-only MI, then we perform several ablation studies, and finally we evaluate if differential privacy (DP) is an effective mitigation.

### 4.1 EXPERIMENTAL SETTING

We perform experiments on four different datasets: three computer vision datasets (GTSRB, CIFAR-10 and CIFAR-100) and one tabular dataset (Purchase-100). We use a ResNet-18 convolutional neural

network model for the vision datasets. We follow the standard training procedure used in prior works (Carlini et al., 2022; Tramèr et al., 2022; Wen et al., 2023), including weight decay and common data augmentations for image datasets, such as random image flips and crops. Each model is trained for 100 epochs, and its training set is constructed by randomly selecting 50% of the original training set.

To instantiate our attack, we pick 500 challenge points at random from the original training set. In the adaptive poisoning stage, we set the poisoning threshold $t_p = 0.15$, the number of OUT models $m = 8$ and the number of maximum poisoning iterations $k_{max} = 6$. In the membership neighborhood stage, we set the neighborhood threshold $t_{nb} = 0.75$, and the size of the neighborhood $|S_{nb}^{(x,y)}| = 64$ samples. Later in Section 4.3, we vary these parameters and explain the rationale behind selecting these values. To construct neighbors in the membership neighborhood, we generate a set of random augmentations for images and select a subset of $64$ augmentations that satisfy Eqn. (1). Finally we test our attack on $64$ target models, trained using the same procedure, including the poisoned set. Among these, 32 serve as IN models, and the remainder as OUT models, in relation to each challenge point. Therefore, the evaluation metrics used for comparison are computed over 32,000 observations.

**Evaluation Metrics.** Consistent with prior work (Carlini et al., 2022; Chen et al., 2022; Tramèr et al., 2022; Wen et al., 2023), our evaluation primarily focuses on True Positive Rate (TPR) at various False Positive Rates (FPRs) namely 0.1%, 1%, 5% and 10%. To provide a comprehensive analysis, we also include the AUC (Area Under the Curve) score of the ROC (Receiver Operating Characteristic) curve and the Membership Inference (MI) accuracy when comparing our attack with prior label-only attacks (Yeom et al., 2018; Choquette-Choo et al., 2021; Li & Zhang, 2021).

## 4.2 CHAMELEON ATTACK IMPROVES LABEL-ONLY MI

We compare our attack against two prior label-only attacks: the Gap attack (Yeom et al., 2018), which predicts any misclassified data point as a non-member and the state-of-the-art Decision-Boundary attack (Choquette-Choo et al., 2021; Li & Zhang, 2021), which uses a sample's distance from the decision boundary to determine its membership status. The Decision-Boundary attack relies on black-box adversarial example attacks (Brendel et al., 2018; Chen et al., 2020). Given a challenge point $(x, y)$, the attack starts from a random point $x'$ for which the model's prediction is *not* label $y$ and walks along the boundary while minimizing the distance to $x$. The perturbation needed to create the adversarial example estimates the distance to the decision boundary, and a sample is considered to be in the training set if the estimated distance is above a threshold, and outside the training set otherwise. Choquette-Choo et al. (2021) showed that their process closely approximates results obtained with a stronger white-box adversarial example technique (Carlini & Wagner, 2017) using $\approx$ 2,500 queries per challenge point. Consequently, we directly compare with the stronger white-box version and show that our attack outperforms even this upper bound.

Table 1: **Comparing Label-only attacks** on GTSRB (G-43), CIFAR-10 (C-10) and CIFAR-100 (C-100) datasets. Our attack achieves high TPR across various FPR values compared to prior attacks.

| | TPR@0.1%FPR | | | TPR@1%FPR | | | TPR@5%FPR | | | TPR@10%FPR | | |
|---|---|---|---|---|---|---|---|---|---|---|---|---|
| **Label-Only Attack** | G-43 | C-10 | C-100 | G-43 | C-10 | C-100 | G-43 | C-10 | C-100 | G-43 | C-10 | C-100 |
| Gap | 0.0% | 0.0% | 0.0% | 0.0% | 0.0% | 0.0% | 0.0% | 0.0% | 0.0% | 0.0% | 0.0% | 0.0% |
| Decision-Boundary | 0.04% | 0.08% | 0.02% | 1.1% | 1.3% | 3.6% | 5.4% | 5.6% | 23.0% | 10.4% | 11.6% | 44.9% |
| **Chameleon (Ours)** | **3.1%** | **8.3%** | **29.6%** | **11.4%** | **22.8%** | **52.5%** | **25.9%** | **34.7%** | **70.9%** | **35.0%** | **42.8%** | **79.4%** |

Table 1 provides a detailed comparison of Chameleon with prior label-only MI attacks. Chameleon shows a significant TPR improvement over all FPRs compared to prior works. In particular, for the case of TPR at $0.1\%$ FPR, prior works achieve TPR values below $0.08\%$, but Chameleon achieves TPR values ranging from $3.1\%$ to $29.6\%$ across the three datasets, marking a substantial improvement ranging from $77.5\times$ to $370\times$. At $1\%$ FPR, the TPR improves by a factor between $10.36\times$ and $17.53\times$. Additionally, our attack consistently surpasses prior methods in terms of AUC and MI accuracy metrics. A detailed comparison can be found in Table 2 (Appendix A.1). Notably, Chameleon is significantly more query-efficient, using only $64$ queries to the target model, compared to the decision-boundary attack, which requires $\approx$ 2,500 queries for the MI test, making our attack approximately $39\times$ more query-efficient.

Furthermore, Chameleon requires adding a relatively low number of poisoned points per challenge point: an average of 3.5 for GTSRB, 1.4 for CIFAR-10, and 0.6 for CIFAR-100 datasets for each

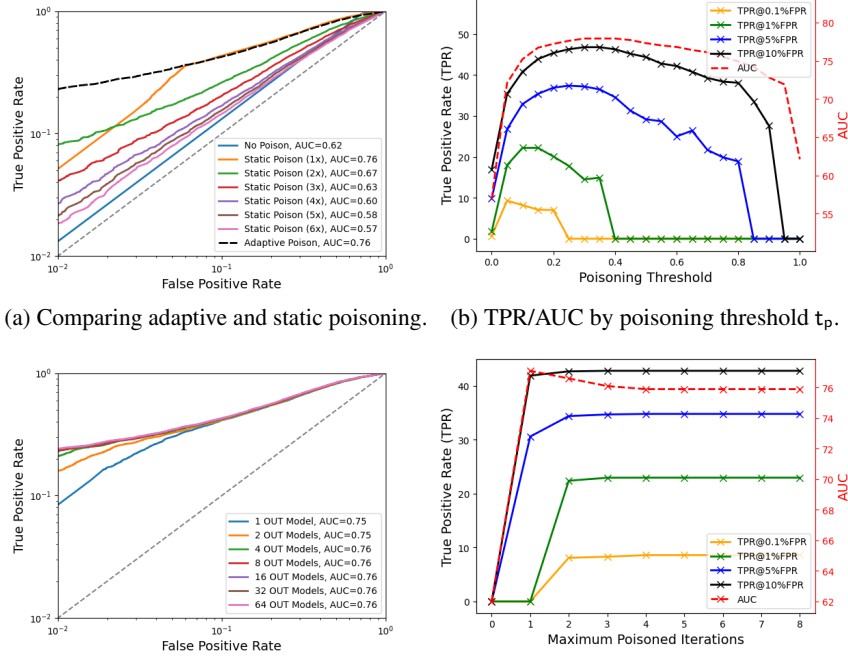

(a) Comparing adaptive and static poisoning.  (b) TPR/AUC by poisoning threshold $t_p$.

(c) TPR/AUC by number of OUT models.  (d) TPR/AUC by number of iterations $k_{max}$.

Figure 5: **Ablations for Adaptive Poisoning stage on CIFAR-10 dataset.** We provide experiments by varying vaious hyperparameters used in the Adaptive Poisoning stage.

challenge point. This results in a minor drop in test accuracy of less than $2\%$, highlighting the stealthiness of our attack. Overall, the results presented in Table 1 show that our adaptive data poisoning strategy significantly amplifies the MI leakage in the label-only scenario while having a marginal effect on the model's test accuracy.

## 4.3 ABLATION STUDIES

**Adaptive Poisoning Stage.** We evaluate the effectiveness of adaptive poisoning and the impact of several parameters.

*i) Comparison to Static Poisoning.* Figure 5a provides a comparison of our adaptive poisoning approach (Algorithm 1) and a static approach where $k$ replicas are added per challenge point. Our approach achieves a TPR@1% FPR of $22.9\%$, while the best static approach among the six versions achieve a TPR@1% FPR of $8.1\%$. The performance improvement of $14.8\%$ in this metric demonstrates the effectiveness of our adaptive poisoning strategy over static poisoning, a strategy used in Truth Serum for confidence-based MI. Additionally, our approach matches the best static approach (with 1 poison) for the AUC metric, achieving an AUC of $76\%$.

*ii) Poisoning Threshold.* Figure 5b illustrates the impact of varying the poisoning threshold $t_p$ (line 7 of Algorithm 1) on the attack's performance. When $t_p = 1$, the OUT model's confidence is at most 1 and Algorithm 1 will not add any poisoned replicas in the training data, but setting $t_p < 1$ allows the algorithm to introduce poisoning. We observe an immediate improvement in the AUC, indicating the benefits of poisoning over the no-poisoning scenario. Similarly, the TPR improves when $t_p$ decreases, as this forces the OUT model's confidence on the true label to be low, increasing the probability of missclassification. However, setting $t_p$ close to 0 leads to overpoisoning, where an overly restrictive $t_p$ forces the algorithm to add a large number of poisoned replicas in the training data, negatively impacting the attack's performance. Therefore, setting $t_p$ between $0.1$ and $0.25$ results in high TPR.

*iii) Number of OUT Models.* Figure 5c shows the impact of the number of OUT models (line 5 of Algorithm 1) on our attack's performance. As we increase the number of OUT models from 1 to 8, the TPR@1% FPR shows a noticeable improvement from $13.1\%$ to $22.9\%$. However, further increasing

the number of OUT models up to $64$ only yields a marginal improvement with TPR increasing to $24.1\%$. Therefore, setting the number of OUT models to $8$ strikes a balance between the attack's success and the computational overhead of training these models.

*iv) Maximum Poisoning Iterations.* In Figure 5d, we observe that increasing the maximum number of poisoning iterations $k_{max}$ (line 2 of Algorithm 1) leads to significant improvements in both the TPR and AUC metrics. When $k_{max} = 0$, it corresponds to the no poisoning case. The TPR@1% FPR and AUC metrics improve as parameter $k_{max}$ increases. However, the TPR stabilizes when $k_{max} \geq 4$, indicating that no more than 4 poisoned replicas are required per challenge point.

**Membership Neighborhood Stage.** Next, we explore the impact of varying the membership neighborhood size and the neighborhood threshold $t_{nb}$ individually. For the neighborhood size, we observe a consistent increase in TPR@1% FPR of 0.6% as we increase the number of queries from $16$ to $64$, beyond which the TPR oscillates. Thus, we set the neighborhood size at 64 queries for our experiments, achieving satisfactory attack success. For the neighborhood threshold parameter $t_{nb}$, we note a decrease in TPR@1% FPR of 6.2% as we increase $t_{nb}$ from 0.25 to 1.75. This aligns with our intuition, that setting $t_{nb}$ to a smaller value prompts our algorithm to select close neighbors, which in turn enhances our attack's performance. The details for these results are in Appendix A.2.

## 4.4 Other Data Modalities and Architectures

To show the generality of our attack, we evaluate Chameleon on various model architectures, including ResNet-34, ResNet-50 and VGG-16. We observe a similar trend of high TPR value at various FPRs. Particularly for VGG-16, which has $12\times$ more trainable parameters than ResNet-18, the attack achieves better performance than ResNet-18 across all metrics, suggesting that more complex models tend to be more susceptible to privacy leakage. We also evaluate Chameleon against a tabular dataset (Purchase-100). Once again, we find a similar pattern of consistently high TPR values across diverse FPR thresholds. In fact, the TPR@1% FPR metric reaches an impressive 45.8% when tested on a two-layered neural network. Experimental setup details and results can be found in Appendix A.3.

## 4.5 Does Differential Privacy Mitigate Chameleon ?

Here, we evaluate Chameleon against models trained using a standard differentially private (DP) training algorithm, DP-SGD (Abadi et al., 2016). Our evaluation covers a broad spectrum of privacy parameters, but here we highlight results on $\epsilon = \{\infty, 100, 4\}$, which represent no privacy, a loose level of privacy, and a strict level of privacy, respectively. At $\epsilon$ as high as 100, we observe a decline in Chameleon's performance, with TPR@1% FPR decreasing from 22.6% (at $\epsilon = \infty$) to 6.1%. In the case of an even stricter $\epsilon = 4$, we observe that TPR@1% FPR becomes 0%, making our attack ineffective. However, it is also important to note that the model's accuracy also degrades significantly, plummeting from 84.3% at $\epsilon = \infty$ to 49.4% at $\epsilon = 4$, causing a substantial 34.9% decrease in accuracy. This trade-off shows that while DP serves as a powerful defense, it does come at the expense of model utility. More results using a wider range of $\epsilon$ values can be found in Appendix A.4.

## 5 Discussion and Conclusion

In this work we propose a new attack that successfully amplifies Membership Inference leakage in the Label-Only setting. Our attack leverages a novel adaptive poisoning and querying strategy, surpassing the effectiveness of prior label-only attacks. Furthermore, we investigate the viability of Differential Privacy as a defense against our attack, considering its impact on model utility. Finally, we offer a theoretical analysis providing insights on the impact of data poisoning on MI leakage. We demonstrated that Chameleon achieves impressive performance in our experiments, mainly due to the adaptive poisoning strategy we design. While poisoning is a core component of our approach, and allows us to enhance the privacy leakage, it also imposes additional burden on the adversary to mount the poisoning attack. One important remaining open problem in label-only MI attacks is how to operate effectively in the low False Positive Rate (FPR) scenario without the assistance of poisoning. Additionally, our poisoning strategy requires training shadow models. Though our approach generally involves a low number of shadow models, any training operation is inherently expensive and adds computational complexity. An interesting direction for future work is the design of poisoning strategies for label-only membership inference that do not require shadow model training.

## ETHICS STATEMENT

Our work introduces a new membership inference methodology, that could in principle be used to exacerbate the privacy risks to individuals' private data when included in machine learning training sets. The main reason motivating us to investigate this type of attacks is to shed light on the extent of the risks, and ensure practitioners are aware of them. In comparison with previous works investigating label-only poisoning-enhanced membership inference attacks, we show that the adversary can manage to reliably compromise the privacy of a significant fraction of the training data. We argue that this type of privacy risk should be seriously taken into account and encourage practitioners to adopt private training procedures, such as differentially private training, to minimize it.

## ACKNOWLEDGEMENTS

We thank Sushant Agarwal and John Abascal for helpful discussions. Alina Oprea was supported by NSF awards CNS-2120603 and CNS-2247484. Jonathan Ullman was supported by NSF awards CNS-2120603, CNS-2232692, and CNS-2247484.

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

## A  ADDITIONAL EXPERIMENTS

In this appendix we include additional results of our experiments with the Chameleon attack.

### A.1  AUC AND MI ACCURACY METRICS

We report here a comparison between our attack and the existing state of the art label-only MI attacks, Gap (Yeom et al., 2018) and Decision-boundary (Choquette-Choo et al., 2021), on aggregate performance metrics: the AUC (Area Under the Curve) score of the ROC (Receiver Operating Characteristic) curve and the average accuracy. Table 2 shows the values of these metrics on the three image datasets used in previous evaluations. Interestingly, Chameleon achieves superior average values in all tested scenarios.

Table 2: **Comparison of Label-only attacks on AUC and Membership Inference (MI) Accuracy metric** for GTSRB, CIFAR-10 and CIFAR-100 datasets. Our attack uses a combination of adaptive poisoning, training shadow models (SMs) and careful selection of multiple queries (MQs) to outperform prior attacks.

| Label-Only Attack | Poison | SMs | MQs | AUC | | | MI Accuracy | | |
|---|---|---|---|---|---|---|---|---|---|
| | | | | GTSRB | CIFAR-10 | CIFAR-100 | GTSRB | CIFAR-10 | CIFAR-100 |
| Gap | ○ | ○ | ○ | 50.6% | 57.7% | 73.8% | 50.6% | 57.7% | 73.8% |
| Decision-Boundary | ○ | ● | ● | 51.5% | 62.8% | 84.9% | 51.3% | 62.4% | 81.1% |
| Chameleon (Ours) | ● | ● | ● | **71.9%** | **76.3%** | **92.6%** | **65.2%** | **68.5%** | **85.2%** |

### A.2  MEMBERSHIP NEIGHBORHOOD STAGE

We analyze the impact on our attack's performance by varying the size of the membership neighborhood set and the neighborhood threshold $t_{nb}$ individually. Recall that the membership neighborhood for a challenge point is designed to compute a proxy score for our distinguishing test, which is achieved by finding candidates that satisfy Equation (1), described in Section 3.2.

*i) Size of Membership Neighborhood.* In Figure 6a, we show the impact of the size of membership neighborhood on the attack success. We observe that as the size of the neighborhood increases from 16 to 64 samples, the TPR@1% FPR and AUC of our attack improves by 0.6% and 3% respectively. However, further increasing the neighborhood size beyond 64 samples does not significantly improve the TPR and AUC, as indicated by the oscillating values. Therefore, setting the membership neighborhood size to 64 samples or larger should provide a satisfactory level of attack performance.

*ii) Neighborhood threshold.* We now vary the neighborhood threshold $t_{nb}$ to observe its impact on the attack's performance. The neighborhood threshold $t_{nb}$ determines the selection of neighbors for the challenge point. In Figure 6b, we observe that setting $t_{nb}$ to 0.25 results in the highest TPR@1% FPR of 23.5%, while increasing $t_{nb}$ to 1.75 decreases the TPR to 17.3%. This aligns with our intuition that

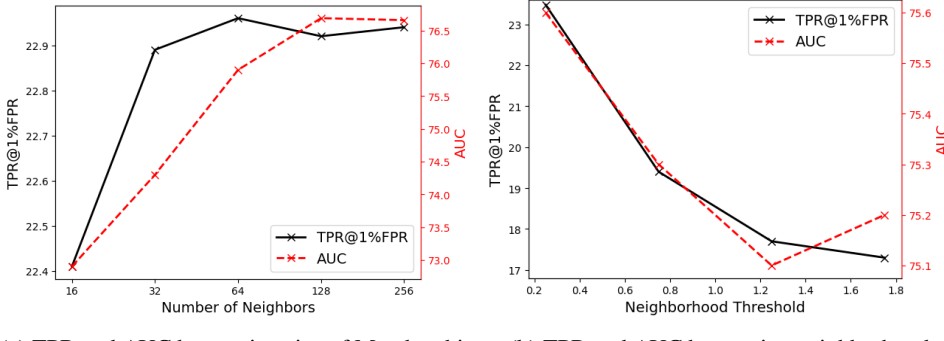

(a) TPR and AUC by varying size of Membership Neighborhood.

(b) TPR and AUC by varying neighborhood threshold $t_{nb}$.

Figure 6: **Ablations for Membership Neighborhood stage on CIFAR-10 dataset.** We provide experiments on two hyperparameters used in this stage: size of the membership neighborhood, and membership threshold

systematically including samples in the membership neighborhood that are closer to the challenge point improves the effectiveness of our attack compared to choosing distant samples or using arbitrary random augmentations. Additionally, we observe a decrease in the AUC score when introducing distant samples in the membership neighborhood. However, this decrease is relatively more robust compared to the TPR metric.

## A.3 DATA MODALITIES AND ARCHITECTURES

Table 3: Effectiveness of Chameleon on Purchase-100 (P-100) and CIFAR-10 (C-10) datasets over various model architectures. Our attack achieves high TPR values across various model architectures.

| Modality | Dataset | Model Type | True Positive Rate (TPR) | | | | AUC | MI Accuracy |
| | | | @0.1%FPR | @1%FPR | @5%FPR | @10%FPR | | |
|---|---|---|---|---|---|---|---|---|
| Tabular | P-100 | 1-NN | 8.7% | 34.8% | 62.2% | 72.9% | 92.0% | 83.8% |
| | | 2-NN | 18.3% | 45.8% | 76.9% | 88.8% | 95.9% | 89.6% |
| Image | C-10 | ResNet-18 | 8.2% | 22.8% | 34.7% | 42.8% | 76.2% | 68.5% |
| | | ResNet-34 | 9.1% | 24.6% | 35.4% | 43.1% | 76.6% | 69.2% |
| | | ResNet-50 | 7.4% | 23.3% | 33.9% | 41.6% | 76.3% | 69.8% |
| | | VGG-16 | 8.6% | 29.4% | 49.1% | 59.3% | 78.6% | 75.4% |

We evaluate here the behavior of our attack on different data modalities, model sizes and architectures.

For our experiments on tabular data, we use the Purchase-100 dataset, with a setup similar to Choquette-Choo et al. (2021): a one-hidden-layer neural network, with 128 internal nodes, trained for 100 epochs. We do not use any augmentation during training, and we construct neighborhood candidates by flipping binary values according to Bernoulli noise with probability 2.5%. Here, we apply the same experimental settings as detailed in Section 4.1, with 500 challenges points and $m = 8$, and a slightly lower $t_p = 0.1$. The first row of Table 3 reports the results of the attack on this modality. Interestingly, despite the limited size of the model, we observe a high AUC and relatively high TPR at 1% FPR.

To observe the effect of Chameleon for different model sizes we experimented with scaling the dimension of both the feed-forward network used for Purchase-100 and the base ResNet model used throughout Section 4. For the feed-forward model we added a single layer, which was already sufficient to achieve perfect training set accuracy, due to the limited complexity of the classification task. On CIFAR-10, we compared the results on ResNet 18, 34 and 50, increasing the number of trainable parameters from 11 million to 23 million. For the larger models we increased the training epochs from 100 to 125. Finally, we considered a VGG-16 (Simonyan & Zisserman, 2015) architecture pre-trained on ImageNet, which we fine-tuned for 70 epochs. This is a considerably larger model, with roughly 138 million parameters.

We observe generally similar attack performance on the ResNet models, as shown in Table 3. The largest model for both modalities, instead, showed significantly higher TPR values at low false positive rates such as 0.1% and 1%. This trend can be attributed to the tendency of larger models to memorize the training data with greater ease.

### A.4 DIFFERENTIAL PRIVACY

Table 4: TPR at various FPR values for our Chameleon attack when models are trained using DP-SGD on CIFAR-10 dataset. Differential Privacy significantly mitigates the impact of our attack but also adversely impacts the model's accuracy.

| Privacy Budget | Model Accuracy | TPR@1%FPR | TPR@5%FPR | TPR@10%FPR | AUC | MI Accuracy |
|---|---|---|---|---|---|---|
| $\epsilon = \infty$ (No DP) | 84.3% | 22.6% | 34.8% | 42.9% | 76.8% | 69.3% |
| $\epsilon = 100$ | 57.6% | 6.1% | 14.8% | 26.7% | 61.8% | 59.7% |
| $\epsilon = 50$ | 56.8% | 0.0% | 11.7% | 21.1% | 58.9% | 57.9% |
| $\epsilon = 32$ | 57.2% | 0.0% | 9.9% | 18.9% | 58.8% | 58.3% |
| $\epsilon = 16$ | 56.4% | 0.0% | 0.0% | 14.2% | 55.8% | 56.0% |
| $\epsilon = 8$ | 54.8% | 0.0% | 0.0% | 12.6% | 53.7% | 53.7% |
| $\epsilon = 4$ | 49.4% | 0.0% | 0.0% | 11.1% | 52.4% | 52.6% |

We evaluate the resilience of our Chameleon attack against models trained using DP-SGD (Abadi et al., 2016). We use PyTorch's differential privacy library, Opacus (Yousefpour et al., 2021), to train our models. The privacy parameters are configured with $\epsilon$ values of $\{4, 8, 16, 32, 50, 100, \infty\}$ and $\delta = 10^{-5}$, alongside a clipping norm of $C = 5$. Our training procedure aligns with that of previous works (Kurakin et al., 2022; De et al., 2022), which involve replacing Batch Normalization with Group Normalization with group size set to $G = 16$ and the omission of data augmentations, which have been observed to reduce model utility when trained with DP. In Table 4, we observe that as $\epsilon$ decreases, our attack success also degrades showing us that DP is effective at mitigating our attack. However, we also observe that the accuracy of the model plummets with decrease in $\epsilon$. Thus, DP can be used as a defense strategy, but comes at a high expense in model utility.

## B ANALYSIS OF LABEL-ONLY MI UNDER POISONING

Let $\mathcal{D}$ denote the distribution from which $n$ samples $z_1, \ldots, z_n$ are sampled in an iid manner. For classification based tasks, a sample is defined as $z_i = (x_i, y_i)$ where $x_i$ denotes the input vector and $y_i$ denotes the class label. We assume binary membership inference variables $m_1, \ldots, m_n$ that are drawn independently with probability $\Pr(m_i = 1) = \lambda$, where samples with $m_i = 1$ are a part of the training set. We model the training algorithm as a random process such that the posterior distribution of the model parameters given the training data $\theta | z_1, \ldots, z_n, m_1, \ldots, m_n$ satisfies

$$\Pr(\theta | z_1, \ldots, z_n, m_1, \ldots, m_n) \propto e^{-\frac{1}{\tau} \sum_{i=1}^{n} m_i L(\theta, z_i)} \tag{2}$$

where $\tau$ and $L$ denote the temperature parameter and the loss functions respectively. Parameter $\tau = 1$ corresponds to the case of the Bayesian posterior, $\tau \to 0$ the case of MAP (Maximum A Posteriori) inference and a small $\tau$ denotes the case of averaged SGD. This assumption on the posterior distribution of the model parameters have also been made in prior membership inference works such as Sablayrolles et al. (2019) and Ye et al. (2022). The prior on $\theta$ is assumed to be uniform.

Without loss of generality, let us analyze the case of $z_1 = (x_1, y_1)$, for a multi-class classification task. Let $C$ denote the total number of classes in the classification task. We introduce poisoning by creating a poisoned dataset $\mathsf{D_p}$ which contains $k$ poisoned replicas of $z_1^p = (x_1, y_1^p)$, where $y_1^p \neq y_1$. Note that, all poisoned replicas have the same poisoned label $y_1^p$ that is distinct from the true label. The posterior in Equation (2) can then be re-written as:

$$\Pr(\theta_p | z_1, \ldots, z_n, m_1, \ldots, m_n, \mathsf{D_p}) \propto e^{-\frac{1}{\tau} \sum_{i=1}^{n} m_i L(\theta_p, z_i) - \frac{k}{\tau} L(\theta_p, z_1^p)} \tag{3}$$

where term $\frac{k}{\tau} L(\theta_p, z_1^p)$ denotes the sum over all the loss terms introduced by $k$ poisoned replicas. Furthermore, we gather information about other samples and their memberships in set $T = \{z_2, \ldots, z_n, m_2, \ldots, m_n\}$. We assume the loss function $L$ to be a 0-1 loss function, so that we can perform a concrete analysis of the loss in Equation (3).

**Assumptions.** We now explicitly list the set of assumptions that will be utilized to design and analyze our optimal attack.

- The posterior distribution of the model parameters given the poisoned dataset satisfies Equation (3), where the loss function $L$ is assumed to be a 0-1 loss function.

- The prior on the model parameters $\theta$ follows a uniform distribution.

- The model parameter selection for classifying challenge point $z_1$ is only dependent on $z_1$, its membership $m_1$ and the poisoned dataset $D_p$. We make this assumption based on the findings from Tramèr et al. (2022), where empirical observations revealed that multiple *poisoned models* aimed at inferring the membership of $z_1$ had very similar logit (scaled confidence) scores for challenge point $z_1$. This observation implied that a poisoned model's prediction on $z_1$ was largely influenced only by the presence/absence of the original point $z_1$ and the poisoned dataset $D_p$.

**Poisoning impact on challenge point classification.** Recall that we are interested in analyzing how the addition of $k$ poisoned replicas influences the correct classification of the challenge point $z_1$ as label $y_1$, considering whether $z_1$ is a member or a non-member. Towards this, we define the event $\theta_p(z_1) = y_1$ as selecting a parameter $\theta_p$ that correctly classifies $z_1$ as $y_1$. More formally, we can write the probability of correct classification as follows:

**Theorem B.1.** *Given the sample $z_1$, binary membership variable $m_1$, poisoned dataset $D_p$ and the remaining training set $T$.*

$$\Pr(\theta_p(z_1) = y_1 | z_1, m_1, T, \mathsf{D_p}) = \frac{e^{-k/\tau}}{e^{-m_1/\tau} + e^{-k/\tau} + (C-2)e^{-(k+m_1)/\tau}} \tag{4}$$

*Proof.* Let us first consider the OUT case when $z_1$ is not a part of the training set, i.e. $m_1 = 0$. Formally we can write the probability of selecting a parameter that results in classification of sample $z_1$ as label $y_1$ as follows:

$$\Pr(\theta_p(z_1) = y_1 | z_1, m_1 = 0, T, \mathsf{D_p})$$

Based on our assumption that under the presence of poisoning, the model parameter selection for classification of $z_1$ depends only on $z_1$, $m_1$ and $\mathsf{D_p}$, we can write

$$\Pr(\theta_p(z_1) = y_1 | z_1, m_1 = 0, T, \mathsf{D_p}) = \Pr(\theta_p(z_1) = y_1 | z_1, m_1 = 0, \mathsf{D_p}) \tag{5}$$

Subsequently, we can use Equation (3) to reformulate Equation (5) as follows:

$$\Pr(\theta_p(z_1) = y_1 | z_1, m_1 = 0, \mathsf{D_p}) = \frac{e^{-k/\tau}}{e^0 + (C-1)e^{-k/\tau}} \tag{6}$$

In this equation, the numerator $e^{-k/\tau}$ denotes the outcome where a parameter $\theta_p$ is chosen resulting in classification of $z_1$ as $y_1$. Similarly, the terms $e^0$ and $(C-1)e^{-k/\tau}$ in the denominator represent the outcomes where a parameter is selected such that it classifies $z_1$ as $y_1^p$ and any other label except $y_1^p$, respectively.

Similar to Equation (6), we can formulate an equation for the IN case as follows:

$$\Pr(\theta_p(z_1) = y_1 | z_1, m_1 = 1, \mathsf{D_p}) = \frac{e^{-k/\tau}}{e^{-1/\tau} + e^{-k/\tau} + (C-2)e^{-(k+1)/\tau}} \tag{7}$$

Similar to Equation (6), the numerator $e^{-k/\tau}$ denotes the outcome that classifies sample $z_1$ as $y_1$. The terms $e^{-1/\tau}$ and $e^{-k/\tau}$ in the denominator represent the outcomes when sample $z_1$ is classified as $y_1^p$ and $y_1$ respectively. Term $(C-2)e^{-(k+1)/\tau}$ denotes the sum over all outcomes where sample $z_1$ is classified as any label except $y_1$ and $y_1^p$. Now, by combining Equation (6) and Equation (7), we arrive at a unified expression represented by Equation (4). □

**Optimal Attack** Our goal now is to formulate an *optimal* attack in the label-only setting that maximizes the TPR value at fixed FPR of x%, when $k$ poisoned replicas of $z_1^p$ are introduced into the training set, adhering to the list of assumptions defined earlier.

We define two events:

- If $\theta(z_1) = y_1$, we say $z_1$ is "IN" the training set with probability $p_0$, else we say "OUT" with probability $1 - p_0$.

- If $\theta(z_1) \neq y_1$, we say $z_1$ is "IN" the training set with probability $p_1$, else we say "OUT" with probability $1 - p_1$.

We can then compute the maximum TPR as follows:

**Theorem B.2.** *Given sample $z_1$ and a training dataset that includes $k$ poisoned replicas of $z_1$. The maximum TPR at $x\%$ FPR is given as*

$$x' \times \frac{(C-1) + e^{k/\tau}}{e^{(k-1)/\tau} + (C-2)e^{-1/\tau} + 1} - p \times \frac{e^{k/\tau} - e^{(k-1)/\tau} + (C-2)(1 - e^{-1/\tau})}{e^{(k-1)/\tau} + (C-2)e^{-1/\tau} + 1}$$

*where probability $p = max\left(0, \frac{x'(C-1) + x'e^{k/\tau} - 1}{(C-2) + e^{k/\tau}}\right)$ and $x' = x/100$.*

*Proof.* Let $x' = x/100$. We can start by writing the equation for FPR as:

$$\Pr(\text{"IN"} \mid z_1, m_1 = 0, T, \mathsf{D_p}) = x'$$

We can expand the left-hand side of the equation as follows:

$$\Pr(\text{"IN"} \mid z_1, m_1 = 0, T, \mathsf{D_p}) = \Pr(\text{"IN"} \mid \theta_p(z_1) = y_1) \times \Pr(\theta_p(z_1) = y_1 \mid z_1, m_1 = 0, T, \mathsf{D_p})$$
$$+ \Pr(\text{"IN"} \mid \theta_p(z_1) \neq y_1) \times \Pr(\theta_p(z_1) \neq y_1 \mid z_1, m_1 = 0, T, \mathsf{D_p})$$
$$= p_0 \times \frac{1}{e^{k/\tau} + (C-1)} + p_1 \times \frac{e^{k/\tau} + (C-2)}{e^{k/\tau} + (C-1)}$$

At a fixed FPR $x'$, the above equation can then be re-written as:

$$p_0 = x' \times ((C-1) + e^{k/\tau}) - p_1 \times ((C-2) + e^{k/\tau}) \tag{8}$$

We also know that the following inequalities hold $0 \leq p_0, \ p_1 \leq 1$. By substituting $p_0$ as a function of $p_1$ from Equation (8), we get:

$$max\left(0, \frac{x'(C-1) + x'e^{k/\tau} - 1}{(C-2) + e^{k/\tau}}\right) \leq p_1 \leq min\left(1, \frac{x'(C-1) + x'e^{k/\tau}}{(C-2) + e^{k/\tau}}\right)$$

Similar to the FPR equation, we formulate the TPR as follows:

$$\Pr(\text{"IN"} \mid z_1, m_1 = 1, T, \mathsf{D_p}) = \Pr(\text{"IN"} \mid \theta_p(z_1) = y_1) \times \Pr(\theta_p(z_1) = y_1 \mid z_1, m_1 = 1, T, \mathsf{D_p})$$
$$+ \Pr(\text{"IN"} \mid \theta_p(z_1) \neq y_1) \times \Pr(\theta_p(z_1) \neq y_1 \mid z_1, m_1 = 1, T, \mathsf{D_p})$$
$$= p_0 \times \frac{1}{e^{(k-1)/\tau} + (C-2)e^{-1/\tau} + 1} + p_1 \times \frac{e^{(k-1)/\tau} + (C-2)e^{-1/\tau}}{e^{(k-1)/\tau} + (C-2)e^{-1/\tau} + 1}$$

We substitute Equation (8) into the above equation and get:

$$= x' \times \frac{(C-1) + e^{k/\tau}}{e^{(k-1)/\tau} + (C-2)e^{-1/\tau} + 1} - p_1 \times \frac{e^{k/\tau} - e^{(k-1)/\tau} + (C-2)(1 - e^{-1/\tau})}{e^{(k-1)/\tau} + (C-2)e^{-1/\tau} + 1} \tag{9}$$

The goal is to maximize the above TPR equation when FPR $= x'$. We can then write Equation (9) as a constrained optimization problem as follows:

$$\max_{p_1} \quad x' \times \frac{(C-1) + e^{k/\tau}}{e^{(k-1)/\tau} + (C-2)e^{-1/\tau} + 1} - p_1 \times \frac{e^{k/\tau} - e^{(k-1)/\tau} + (C-2)(1 - e^{-1/\tau})}{e^{(k-1)/\tau} + (C-2)e^{-1/\tau} + 1}$$

$$(10)$$

$$\text{s.t.} \quad max\left(0, \frac{x'(C-1) + x'e^{k/\tau} - 1}{(C-2) + e^{k/\tau}}\right) \le p_1 \le min\left(1, \frac{x'(C-1) + x'e^{k/\tau}}{(C-2) + e^{k/\tau}}\right)$$

In Equation (10), we observe that the first term is a constant and the coefficient of $p_1$ is positive. Consequently, we must set $p_1$ to its minimum possible value in order to maximize the TPR value. Thus the TPR equation can be re-written as:

$$= x' \times \frac{(C-1) + e^{k/\tau}}{e^{(k-1)/\tau} + (C-2)e^{-1/\tau} + 1} - p \times \frac{e^{k/\tau} - e^{(k-1)/\tau} + (C-2)(1 - e^{-1/\tau})}{e^{(k-1)/\tau} + (C-2)e^{-1/\tau} + 1} \quad (11)$$

where probability $p = max\left(0, \frac{x'(C-1) + x'e^{k/\tau} - 1}{(C-2) + e^{k/\tau}}\right)$.

$\square$

As previously shown in Figure 4 (Section 3.3), we plot the TPR as a function of the number of poisoned replicas for our setting using Theorem B.2. We set the temperature parameter to a small value $\tau = 0.5$, the number of classes $C = 10$. We fix the FPR to $5\%$ and plot our theoretical attack. In order to validate the similarity in behavior between our theoretical model and practical scenario, we run the static version of our label-only attack where we add $k$ poisoned replicas for a challenge point in CIFAR-10 dataset. We observe that the TPR improves with increase with introduction of poisoning and then decreases as the number of poisoned replicas gets higher. Note that, the assumptions made in our theoretical analysis do not hold in the absence of poisoning ($k = 0$). Hence, we see a discrepancy between the practical and theoretical attack at $k = 0$.

## C  HANDLING MULTIPLE CHALLENGE POINTS

**Adaptive Poisoning Strategy.**   We now explore how to extend Algorithm 1 from Section 3.1 to infer membership on a set of $n$ challenge points. One straightforward extension is applying Algorithm 1 separately to each of the $n$ challenge points, but there are several drawbacks to this approach. First, the number of shadow models grows proportionally to the number of challenge points $n$, rapidly increasing the cost and impracticality of our method. Second, this method would introduce poisoned replicas by analyzing each challenge point independently, overlooking the potential influence of the presence or absence of other challenge points and their associated poisoned replicas in the training data.

Consequently, we propose Algorithm 2, which operates over multiple challenge points by training a fixed number of shadow models. In Algorithm 2 (Step 3), we start with constructing $2m$ subsets by randomly sampling half of the original training set. These subsets provide various combinations of challenge points, depending on their presence or absence in the subset, helping us tackle the second drawback. We then use these $2m$ subsets to iteratively construct the poisoned set $\mathsf{D_p}$ and train $2m$ shadow models per iteration (Steps 4-16). Thus, the total number of shadow models trained over the course of Algorithm 2 is $2(\mathsf{k_{max}} + 1)m$, where $m$ and $\mathsf{k_{max}}$ are hyperparameters chosen by the attacker and not dependent on the number of challenge points $n$. In fact, the cost of Algorithm 2 is only a constant factor $2\times$ higher than Algorithm 1 that was originally designed for a single challenge point. Later in Appendix D we analyze the impact on our attack's success by varying hyperparameters $m$ and $\mathsf{k_{max}}$.

**Membership Neighborhood.**   Recall that, to construct the membership neighborhood for each challenge point, the attacker needed both IN and OUT shadow models specific to that challenge

---

**Algorithm 2** Adaptive poisoning strategy on a set of challenge points

---

**Input:** Dataset $D_{adv}$, set of challenge points $S_c = \{(x_1, y_1), \ldots, (x_n, y_n)\} \subset D_{adv}$, set of poisoned points $S_p = \{(x_1, y_1'), \ldots, (x_n, y_n')\}$ such that $\forall_{i \in [1,n]} y_i' \neq y_i$, poison threshold $t_p$ and maximum iterations $k_{max}$, $m$ number of OUT models to train.

1: Set of poisoned replicas counters $S_k = \{k_1 = 0, \ldots, k_n = 0\}$.

2: Initialize a set $\{b_1 = 0, \ldots, b_n = 0\}$ to keep track of break condition.

3: Construct subsets $D_i$, with $i \in [1, 2m]$, by randomly sampling half of $D_{adv}$.

4: **For** $k = 0, \ldots, k_{max}$ **do:**

5:      Construct poisoned dataset $D_p$ such that $\forall_{i \in [1,n]}$ $D_p$ contains $k_i$ replicas of $(x_i, y_i')$.

6:      Train $2m$ models $\{\theta_1, \ldots, \theta_{2m}\}$ on dataset $D_p \cup D_{i \in [1,2m]}$

7:      **For** $i = 1, \ldots, n$ **do:**

8:          OUT $\leftarrow$ subset of $m$ models whose training data did not include challenge point $i$

9:          Query $x_i$ on the OUT models and obtain model confidences $\{c_1^{y_i}, \ldots, c_m^{y_i}\}$

10:         Compute the mean $\mu_i = \frac{\sum_{j=1}^{m} c_j^{y_i}}{m}$.

11:           **If** $\mu_i \leq t_p$**:**

12:               Set $b_i = 1$.

13:           **Else:**

14:               $k_i = k_i + 1$.

15:      **If** $\sum_{i=1}^{n} b_i = n$ **:**

16:          **break**

**Output:** Set of number of poisoned replicas $S_k$.

---

point. However by design of Algorithm 2, we can now repurpose the $2m$ models ($m$ IN and $m$ OUT) trained during the adaptive poisoning stage to build the neighborhood. Consequently, there is no need to train any additional shadow models, making this stage very efficient.

## D  ATTACK SUCCESS AND COST ANALYSIS

We perform a detailed analysis on the computational cost of our attack and its implications on our attack's success. Recall in Appendix C, we determined the total number of shadow models to be $2(k_{max} + 1)m$, where $m$ and $k_{max}$ denote the hyperparameters in Algorithm 2. In Table 5, we vary these parameters and observe their effects on our attack's success and the number of shadow models trained upon the algorithm's completion. Each entry in Table 5 represents a tuple indicating TPR@1%FPR and the total shadow models trained, respectively. The results are presented for $k_{max} \leq 4$, as Algorithm 2 terminates early (at Step 15) when $k_{max} \geq 5$.

Table 5: Evaluation of attack success and computational cost for our Chameleon attack on CIFAR-10 dataset by varying hyperparameters $m$ (number of OUT models) and $k_{max}$ (maximum iterations) given in Algorithm 2. Each entry is presented as a tuple, indicating TPR@1%FPR and the total number of (ResNet-18) shadow models trained at the completion of Algorithm 2.

| CIFAR-10 | $m = 1$ | $m = 2$ | $m = 4$ | $m = 8$ |
|---|---|---|---|---|
| $k_{max} = 1$ | (0%, 2) | (0%, 4) | (1.1%, 8) | (1.1%, 16) |
| $k_{max} = 2$ | (13.2%, 6) | (15.6%, 12) | (20.7%, 24) | (22.4%, 48) |
| $k_{max} = 3$ | (13.2%, 8) | (15.8%, 16) | (20.9%, 32) | (22.8%, 64) |
| $k_{max} = 4$ | (13.6%, 10) | (15.9%, 20) | (21.1%, 40) | (22.9%, 80) |

We observe that our attack attains a notable TPR of 22.9% when setting $m = 8$ and $k = 4$, albeit at the expense of training 80 shadow models. However, for the attack configuration with $m = 2$ and $k = 3$, Chameleon still achieves a high TPR of 15.8% while requiring only 16 shadow models. This computationally constrained variant of our attack still demonstrates a TPR improvement of $12.1\times$

over the state-of-the-art Decision Boundary attacks. Consequently, in computationally restrictive scenarios, a practical guideline would be to set $m = 2$ and $k = 3$.

Interestingly, even our computationally expensive variant ($m = 8$ and $k = 4$) still trains fewer models than the state-of-the-art confidence-based attacks like Carlini et al. (2022); Tramèr et al. (2022), which typically use 128 shadow models (64 IN and 64 OUT).

We also conduct a cost analysis on the more complex CIFAR-100 dataset, as presented in Table 6. We observe similar TPR improvement of $13.1\times$ over the Decision-Boundary attack, while training as few as 12 shadow models. With the CIFAR-100 dataset, our algorithm terminates even earlier at $k_{max} = 2$, requiring fewer models to be trained.

Table 6: Analyzing Chameleon attack success and computational cost on CIFAR-100, varying hyperparameters $m$ and $k_{max}$ (Algorithm 2). Entries represent TPR@1%FPR and the total number of (ReseNet-18) shadow models trained.

| CIFAR-100 | $m = 1$ | $m = 2$ | $m = 4$ | $m = 8$ |
|---|---|---|---|---|
| $k_{max} = 1$ | (33.8%, 2) | (43.7%, 4) | (50.3%, 8) | (51.1%, 16) |
| $k_{max} = 2$ | (37.2%, 6) | (47.2%, 12) | (50.7%, 24) | (52.5%, 48) |

*Query Complexity:* Though prior Decision-Boundary attacks (Choquette-Choo et al., 2021; Li & Zhang, 2021) do not train any shadow models, they do require a large number of queries (typically 2,500+ queries) to be made to the target model per challenge point. On the contrary, after training $2(k_{max} + 1)m$ shadow models for a set of $n$ challenge points, our attack only requires atmost 64 queries per challenge point to the target model. This makes our attack $> 39\times$ more query-efficient than prior attack.

*Running Time:* We present the average running time for both attacks, on a machine with an AMD Threadripper 5955WX and a single NVIDIA RTX 4090. We run the Decision-Boundary (DB) attack using the parameters provided by the code[1] in Li & Zhang (2021) on CIFAR-10 dataset. It takes 29.1 minutes to run the attack on 500 challenge points while achieving a TPR of 1.1% (@1%FPR).

For our attack, training each ResNet-18 shadow model requires about 80 seconds. That translates to 106.7 minutes of training time for the expensive configuration of m=8 and k=4 but achieves a substantial TPR improvement of $20.8\times$ compared to the DB attack. Conversely, the computational restricted version of our attack (m=2 and k=3) takes only 21.3 minutes of training time while still achieving a significant TPR improvement of $12.1\times$ over the DB attack.

Note that, the membership neighborhood stage in our attack requires only access to the non-poisoned shadow models, allowing it to run in parallel as soon as the first iteration (Step 4, Algorithm 2) of the adaptive poisoning stage concludes, incurring no additional time overhead. Our querying phase takes approximately a second to query 500 challenge points and their respective neighborhoods (each of size 64).

## E  PRIVACY GAME

We consider a privacy game, where the attacker aims to guess if a challenge point $(x, y) \sim \mathcal{D}$ is present in the challenger's training dataset $D_{tr}$. The game between the challenger $\mathcal{C}$ and attacker $\mathcal{A}$ proceeds as follows:

1:  $\mathcal{C}$ samples training data $D_{tr}$ from the underlying distribution $\mathcal{D}$.
2:  $\mathcal{C}$ randomly selects $b \in \{0, 1\}$. If $b = 0$, $\mathcal{C}$ samples a point $(x, y) \sim \mathcal{D}$ uniformly at random, such that $(x, y) \notin D_{tr}$. Else, samples $(x, y)$ from $D_{tr}$ uniformly at random.
3:  $\mathcal{C}$ sends the challenge point $(x, y)$ to $\mathcal{A}$.
4:  $\mathcal{A}$ constructs a poisoned dataset $D_p$ and sends it to $\mathcal{C}$.
5:  $\mathcal{C}$ trains a target model $\theta_t$ on the poisoned dataset $D_{tr} \cup D_p$.
6:  $\mathcal{C}$ gives $\mathcal{A}$ label-only access to the target model $\theta_t$.
7:  $\mathcal{A}$ queries the target model $\theta_t$, guesses a bit $\hat{b}$ and wins if $\hat{b} = b$.

---

[1]https://github.com/zhenglisec/Decision-based-MIA

The challenger $\mathcal{C}$ samples training data $\mathsf{D}_{\mathsf{tr}} \sim \mathcal{D}$ from an underlying data distribution $\mathcal{D}$. The attacker $\mathcal{A}$ has the capability to inject additional poisoned data $\mathsf{D}_{\mathsf{p}}$ into the training data $\mathsf{D}_{\mathsf{tr}}$. The objective of the attacker is to enhance its ability to infer if a specific point $(x, y)$ is present in the training data by interacting with a model trained by challenger $\mathcal{C}$ on data $\mathsf{D}_{\mathsf{tr}} \cup \mathsf{D}_{\mathsf{p}}$. The attacker can only inject $\mathsf{D}_{\mathsf{p}}$ once before the training process begins, and after training, it can only interact with the final trained model to obtain predicted labels. Note that, both the challenger and the attacker have access to the underlying data distribution $\mathcal{D}$, and know the challenge point $(x, y)$ and training algorithm $\mathcal{T}$, similar to prior works (Carlini et al., 2022; Tramèr et al., 2022; Chen et al., 2022; Wen et al., 2023).

