# OpenReview forum: "Chameleon: Increasing Label-Only Membership Leakage with Adaptive Poisoning"
_ICLR.cc/2024/Conference — ICLR 2024 poster_

### Official Review · Reviewer_JKci · 2023-10-29

**Soundness:** 2 fair
**Presentation:** 3 good
**Contribution:** 3 good
**Rating:** 6
**Confidence:** 4

**Summary:**

The paper proposes a data poisoning attack, called Chameleon, to enhance the privacy leakage due to label only membership inference attacks. It first shows that current attacks that aim to enhance privacy leakage via poisoning are not effective in label-only MIA threat model. The attack fails because after poisoning both the IN and OUT models misclassify the target samples. To improve the attack efficacy, Chameleon tailors the number of replicas of poisoning samples for each challenge sample. The paper shows that such poisoning significantly improves label-only MIA accuracy especially at low FPRs.

**Strengths:**

- Chameleon idea is elegant and easy to implement
- Intuition and other aspects of the attack are well explained

**Weaknesses:**

- Chameleon is an expensive attack
- I am not sure how will such attack be useful in practice due to the computations involved
- Some parts of the paper need to improve presentation, e.g., theoretical attack and figure 1

**Questions:**

The attack proposed is very elegant in that it is easy to implement and outperforms prior attacks. Also, the explanation of the attack is  clear and easy to understand. The paper also does a fair job in evaluating their proposed attack. Overall I think this is a good paper, but  I have the following concerns:

Attack computation cost and utility:
- Chameleon is an expensive attack given the number of models one has to train to find the right number of poisoning sample replicas. Can authors discuss the compute cost involved? I didn’t see any discussion in the main paper.
- Given the high computation cost and the fact that modern ML model architectures are generally huge, I wonder where will this attack be useful? Which type of adversaries can afford it? It will be good to clearly discuss these aspects.

Some concerns about the evaluations
- For C100, Chameleon adds on average 0.6 replicas of poisoning samples, which means there are 40% data which need no poisoning. This means MIAs without any poisoning should work well. But this is not reflected in Table 1 results. Clarify.
- Minor: Given that modern ML systems have generally very large and multimodal models, it might be useful to have evaluations on large and/or multimodal models.

Clarity of paper:
- Figure 1 is not readable: I could not understand what it is trying to convey. Please clarify
- Theoretical attack section currently does not clearly explain what is the attack and why this analysis is needed if the same conclusions can be drawn from empirical analysis.

---

> ### Author Response · Authors · 2023-11-15
>
> We thank the reviewer for their insightful comments. We address the main concerns below.  In the paper, we marked the modified text in blue, for ease of identification.
>
> **Attack computation cost and utility:**
>
> While we acknowledge that shadow model training incurs a cost, it is crucial to note that our attack trains a fixed number of shadow models, independent of the number of challenge points. In our experiments for CIFAR-10, inferring membership over 500 challenge points simultaneously, our attack trains a total of 80 shadow models and achieves a 20.8x TPR(@1%FPR) improvement over the Decision-Boundary (DB) attack. Despite the training cost, it still remains notably lower than that of state-of-the-art confidence based membership inference attacks like LiRA (Carlini et al., 2022) and TruthSerum (Tramer et al., 2022), which typically require 128 shadow models for optimal performance.
>
> Additionally, by careful selection of hyperparameters in Algorithm 2, our attack achieves a 12.1x TPR(@1%FPR) improvement over the DB attack while requiring to train only 16 shadow models. In Appendix D, we provide a detailed cost analysis associated with running our attack along with end-to-end running time comparison between ours and DB attack. Interestingly, our computationally constrained variant, using 16 models, nearly matches the runtime of the DB attack while significantly outperforming it in terms of TPR (12.1x improvement).
>
>
> **Evaluation Concerns:**
>
> - **CIFAR-100:** Table 1 displays TPR values at low FPRs, with False Positives in CIFAR-100 coming from 60% of the challenge points that require poisoning to succeed in the MI test. Existing Decision-Boundary attack indeed demonstrates a strong performance (as shown in Table 2) but on average case metrics with AUC and MI accuracy being 84% and 81%, respectively. Notably, these metrics are high because of the 40% of challenge points that succeed the MI test without requiring poisoning.
>
>
> - **Large Models:** We evaluated our attack on a larger model (VGG-16) with 138 million trainable parameters. In this scenario we observed an increase in MI leakage with respect to the smaller ResNet-18 model. Results for this setup are given in Section 4.4 (Page 9) and Table 3 (Appendix A.3, Page 13).
>
> **Clarity:**
>
> - **Figure 1:** The objective of Figure 1 is to show how two distinct challenge points require different numbers of poisoned replicas to create a separation between  the IN and OUT model confidences. We have updated Figure 1 to make it more readable.
>
>
> - **Theoretical/Optimal Attack:** We aim to formulate an optimal attack that maximizes the TPR at a fixed FPR of x%, under a specified list of assumptions (detailed in Appendix B.1).  Our objective of constructing the optimal attack is two-fold:
>
>     - To understand how the maximum attainable TPR (@x%FPR) is affected when we vary the number of poisoned replicas in the training set.
>     - To analyze if our Chameleon attack follows the optimal attack’s behavior or deviates from it under poisoning.
>
> We observe that the maximum attainable TPR improves and then declines as we add more poisoned replicas in the training set  indicating that excessive poisoning in
> the label-only scenario adversely impacts the maximum achievable TPR. A similar behavior is also observed when we run the Chameleon attack on real data showing that it closely aligns with the optimal attack behavior. We have revised Section 3.3 for improved clarity on the optimal attack and its objectives.
>
> We hope our response has addressed the concerns expressed in the review. We will be happy to provide additional clarifications if needed.

---

> > ### Comment · Reviewer_JKci · 2023-11-19
> > **Thanks for the response!**
> >
> > - Time complexity: Thanks for clarifying. The part that talks about optimized algorithm for attack is not visible in the main body of the paper and I think it would be good if authors give Algorithm 2, Appendix C, and Appendix D more visibility in the main body as this is the algorithm that probably will be useful not the Algorithm 1.
> >
> > - Evaluation concern:
> > 1. Cifar100: Is this your hypothesis or you know that all the false positives are from 60% of the points?
> > 2. For larger models, why have you increased number of training epochs? Wouldn't that lead to unnecessary overfitting?
> >
> > - Clarity:
> > 2. Theoretical attack: If I understand correctly, optimal attack should always outperform other attacks, including your attack. Why is this not the case here?
> >
> > - **New concerns**:
> > 1. How do you choose y' when poisoning? Do you randomly pick some label other than true label? What is the impact of choice of y' on performances of your and baseline MIAs? If you haven't already, please add this somewhere in the paper.
> >
> > 2. Evaluation setup:  Given that label only attacks are designed to be more practical, how can attackers have the data from exactly the same distribution of the original training data? This is a strong assumption, and hence, can you provide some evaluations where attacker cannot have data from exactly the same distribution?
> >
> > 3. Finally, I see that you discuss DP as a defense, but it is tough to achieve good privacy-utility trade-offs with DP. So, I was wondering if you can comment on (or even better show) how would these attacks work with simpler defenses, e.g., regularization or per-example gradient clipping?
> >
> > 4. Also, can you provide train and test accuracies of models you are attacking? Before and after poisoning? This is important because if for 500 challenge points, attacker introduces D_p of size 2000 (~10% of training data), and if that leads to poor model performances, such model will never be deployed, and hence, will not be available for attacker to query.
> >
> > 5. If I understand correctly, this is a targeted label-only MIA where you report results only on the set of challenge points. If yes, how have you ensured significance of the results reported? Do you repeat the experiments before reporting results?

---

> > > ### Author Response · Authors · 2023-11-21
> > > **Follow up**
> > >
> > > **Time complexity:** We will update  the paragraph in section 3.3 to reflect the optimized variant of our attack, as suggested by the reviewer.
> > >
> > > **CIFAR-100:** We experimentally verified our intuition over multiple models where we checked the samples that caused 1% FPR for the decision boundary attack. They, indeed, required poisoning for the membership inference attack to be successful.
> > >
> > > **Larger models:**  After 100 epochs, the ResNet-18 model achieved a test accuracy of approximately 83-84%. Meanwhile, ResNet-50, using the same training setup for 100 epochs, showed lower test accuracy in the range of 79-80%. To ensure a fair comparison, we increased the training epochs to 125 for ResNet-50, resulting in a similar test accuracy of 83-84% comparable to the ResNet-18 model.
> > >
> > > **Theoretical attack:**  The optimal attack will outperform  all attacks that follow the same assumptions as the optimal attack. However, real-world scenarios may not always align perfectly with these assumptions, leading to the observed discrepancies in the plot. The primary aim behind constructing the optimal attack was to provide a better understanding on the behavior of the Chameleon attack rather than solely outperforming it.
> > >
> > > **New Concerns:**
> > >
> > > **Q1:** We use the random label flipping strategy for poisoning outlined by Truth Serum (Tramer et al., 2022). Their work showed that there was no substantial advantage in using alternative label flipping methods (like selecting the next most likely class or the least likely class) over assigning a random label different from the original. We will add this reasoning in the revised paper.
> > >
> > > **Q2:** We adhere to the same data distribution assumption that has been followed in the MI literature, which also includes state-of-the-art  Label-Only (LO) attacks that we compare against: Shorki et al., 2017, Yeom et al., 2018, Leino et al., 2020, Choo et al. (LO), 2021, Li et al., 2021 (LO), Carlini et al., 2022, Tramer et al, 2022 and Wen et al., 2023.
> > >
> > > **Q3:** We train our models with weight decay regularization, using standard parameters common in training CIFAR-10 models (weight_decay=5e-4). We also run some preliminary experiments with a more aggressive regularization setting (weight_decay=5e-3). This approach appears to partially mitigate our attack, reducing the TPR at 1% FPR from 22.8% to 5.5% and the AUC from 76.2% to 64.1%. Note that our attack still shows 14x improvement over the decision boundary attack, which achieves a TPR of 0.4% at 1% FPR under the same heavy regularization. However this mitigation comes at the cost of training accuracy of the victim model: we observed a decrease of ~ 6% training accuracy. We will investigate in more details the effect of regularization on our attack. We also note that prior work by Kaya et al., 2020 (https://arxiv.org/abs/2006.05336), discourages the exclusive use of regularization methods to defend against membership inference, as it may provide a false sense of security against more powerful threat models.
> > >
> > > **Q4:** We observed a minor accuracy drop of  less than 2% before and after poisoning across our datasets (when tested on ResNet-18 models) highlighting the stealthiness of our attack. For instance, for the CIFAR-10 dataset, the model's test accuracy marginally decreased from  83.8% before poisoning to 82.4% after.
> > >
> > > **Q5:** We follow the same evaluation methodology as established in prior works  (Carlini et al., 2022; Tramer et al., 2022; Wen et al., 2023), where  we repeat our attack evaluation 64 times and present the average over 64 trials. Each trial involves training a target model on the poisoned dataset and assessing membership inference on the 500 challenge points, which were also chosen randomly in the beginning. The comparison presented in Table 1 against prior works is also averaged over 64 trials to ensure consistency.

---

> > > > ### Comment · Reviewer_JKci · 2023-11-22
> > > > **Thanks for the response!**
> > > >
> > > > Thanks for the detailed response. I think most of my concerns are addressed, except about the assumptions about the adversary's knowledge of distribution of the training data. I understand that prior works do not consider real-world use cases either, but I feel it should be included in newer works as this one.

---

> > > > > ### Author Response · Authors · 2023-11-22
> > > > > **Treating it as a separate problem.**
> > > > >
> > > > > We thank the reviewer for engaging with us in discussing the merits of our work.
> > > > >
> > > > > As a last note, we would like to clarify that the primary aim of this work was to analyze if prior-label only attacks worked in the low FPR regime and consequently propose a better attack that improves upon these works under the existing threat model. Changing the threat model to one that focuses on distribution shifts, as suggested by the reviewer, would require a systematic re-evaluation of existing MI attacks. This evaluation would aim to understand their effectiveness within this new threat model. Following this assessment, if these attacks do not work, the goal would be to develop an attack designed to withstand distribution shifts. We believe this to be a broader problem for MI literature (white-box, black-box and label-only attacks) and deserves to be treated separately.

---

### Official Review · Reviewer_ECEy · 2023-11-05

**Soundness:** 3 good
**Presentation:** 3 good
**Contribution:** 2 fair
**Rating:** 5
**Confidence:** 4

**Summary:**

The key contribution of this paper is to present a poisoning strategy to enhance the success of label-only membership inference attacks. The paper first shows that an existing poisoning regime negatively impacts the label-only attack's success and proposes a new way to calibrate the number of poisoning points to inject. And then, the paper proposes a way to construct shadow models and perform membership inference. In evaluation, the paper demonstrates that poisoning can increase the TPR by an order of magnitude while preserving the model's performance. The paper also analyzes the impact of attack configurations and further tests if (and also shows) DP reduces the attack success.

**Strengths:**

1. The paper presents a new poisoning attack for enhancing label-only MIs.
2. The paper shows the poisoning can increase the attack success by 18x.
3. The paper is well-written

**Weaknesses:**

1. The poisoning seems to be a straightforward adaptation of Tramer et al.
2. The proposed label-only MI seems to be impractical.
3. (Sec 3.3) The claim about "theoretical" attack is unclear.


Overall comments:

I agree it is a nice extension of existing work (Tramer et al.) to label-only settings. At the same time, this attack itself and the poisoning strategy are not surprising. So, my impression of this paper is slightly below the acceptance threshold. But if there are surprising factors that I've missed, I am not willing to fight for rejection.


**[Straightforward Extension]**

Of course, existing poisoning could not work well against an adversary who only observes hard labels. The adversary cannot "exploit" the impacts of poisoning until there is a change in the target's label. If too many are injected, the attacker may not know whether the target is a member. So, in the label-only settings, the key is to calibrate the number of poisoning samples. It is therefore not surprising in Section 3.2 that an "adaptive" poisoning strategy is needed.


**[Practicality of This Poisoning]**

However, I believe that choosing the right threshold $t_p$ is more challenging than shown in this paper. The paper assumes that the adversary can know the "underlying distribution."

But considering that the label-only attacks are for studying the practicality in the "true black-box" settings (e.g., hard-labels), I wonder how well this attack can perform when there's a slight distributional difference between the training data an adversary uses and the victim's. Indirectly, the ablation study shows the proposed label-only attack is a bit sensitive to the choice of a poisoning threshold.

In practical scenarios, when a practitioner wants to check the risks of "practical" label-only membership leakage, the proposed attack may not be a useful one to use.


**[Theoretical Attacks (Sec 3.3)]**

(1) In most cases, the theoretical analysis means the best possible attack that an adversary can perform under a specific attack configuration. But I am not sure whether the paper presents the same.

(2) I am a bit unclear on how the paper theoretically analyzes the impact of poisoning samples on the leakage. It depends on many factors, such as the training data and/or the choice of a model and a training algorithm.

I think the section could a bit mislead readers.

**Questions:**

My questions are in the detailed comments in the weakness section.

**Details Of Ethics Concerns:**

No concern about the ethics.

---

> ### Author Response · Authors · 2023-11-15
>
> We thank the reviewer for the careful reading of the paper, and the detailed comments. We address the main concerns below.  In the paper, we marked the modified text in blue, for ease of identification.
>
> **Straightforward Extension:**
>
> The Truth Serum attack by Tramer et al. does not work in label-only settings, and we had to add several new components to account for this challenging threat model: (1) We introduce a new adaptive poisoning strategy that is critical for the attack's success in label-only settings; (2) We carefully craft queries based on our proposed concept of membership neighborhood to improve the attack’s success. Both of these ideas are novel and have not been used in prior work on membership inference. In more detail, our contributions are as follows:
>
> - **Poisoning Strategy:**
>     - **Single challenge point:** Algorithm 1 is designed to adaptively calibrate the number of  poisoned replicas k such that the label for the given challenge point flips with respect  to the OUT model, creating a separation between IN and OUT models.
>
>     - **Multiple challenge points:** The first crucial observation we make is that each challenge point  requires a different  number of poisoned replicas k to create a separation between the IN and OUT models. In Fig. 5(a), we show that fixing the same k across all challenge points does not yield a high TPR.  In Appendix C, we then discuss that trivially repeating Algorithm 1 for each challenge point makes our attack impractical, as the number of shadow models to be trained scales up with the number of challenge points. Consequently, we propose Algorithm 2, which tackles this drawback and trains a fixed number of shadow models, independent of the number of challenge points.
>
> - **Membership Neighborhood:** We introduce a systematic approach for selecting close neighbors for each challenge point by reusing the shadow models trained in the poisoning stage. With only 64 queries per challenge point, our attack is >39x more query-efficient than the prior Decision Boundary attacks. Figure 6(b) demonstrates how using heuristic augmentations instead of our systematic approach for querying the target model can cause a significant drop in the TPR.
>
> **Theoretical/Optimal Attack** (Appendix B):
>
> The analysis indeed builds an optimal attack that maximizes TPR at a fixed FPR of x% for a given attack configuration. The attack configuration (as detailed in Appendix B) uses the following assumptions:
>
> - The posterior distribution of the model parameters given the training data satisfies Equation (2). This is a common assumption made by prior MI works Sablayrolles et al. (2019) and Ye et al. (2022)  when modeling MI leakage.
> - We also assume the 0-1 loss function is used in order to bound the total loss in  Equation (2).
> - Finally, we assume that under poisoning, the model parameter selection for classifying  a specific challenge point z_1 is solely dependent on z_1,  its membership m_1 and the poisoned dataset D_p. This assumption is based on the findings from TruthSerum (Tramer et al., 2022), where empirical observations revealed that multiple poisoned models aimed at inferring the membership of z_1 had very similar logit (scaled confidence) scores on challenge point z_1. This observation implied that a poisoned model’s prediction on z_1 was largely influenced only by the presence/absence of the original point and the poisoned dataset D_p.
>
> We have also revised Section 3.3 to improve clarity of our optimal attack and its objectives, under the attack configuration mentioned above.
>
> **Practicality of Poisoning:**
>
> We adhere to the same data distribution assumption that has been followed in the MI literature (including prior label-only attacks): Shorki et al., 2017, Yeom et al., 2018, Leino et al., 2020, Choo et al., 2021, Li et al., 2021,  Carlini et al., 2022, Tramer et al, 2022 and Wen et al., 2023.
>
> We hope our response has addressed the main concerns expressed in the review. We will be happy to provide additional clarifications if needed.

---

### Official Review · Reviewer_uopX · 2023-11-05

**Soundness:** 3 good
**Presentation:** 3 good
**Contribution:** 3 good
**Rating:** 5
**Confidence:** 4

**Summary:**

This paper targets at Membership Inference (MI), in which an attacker seeks to determine whether a particular data sample was included in the training dataset of a model. In contrast to the most of work in this area, the paper considers a less favorable setting: the attacker has access only to the predicted label on a queried sample, instead of the confidence level. I think this is an important problem, which should be interesting to the communities of both DP and privacy attack. To address this challenge, the paper proposes a new
attack Chameleon that leverages adaptive data poisoning to achieve better accuracy than the previous work.

**Strengths:**

1. The paper proposes a new attack Chameleon that leverages adaptive data poisoning to achieve better accuracy than the previous work.
2. The paper observes an interesting phenomenon: for different challenge point, the sweet spot of the number of samples needed in the data poisoning is different. The paper also proposes a theory to reflect this phenomenon.
3. Various experiments have shown the advantages of the new method.

**Weaknesses:**

Although the attack and the observation is interesting, I think the paper has the following weak points:

1. Time complexity. Clearly from Algorithm 1, to run the adaptive poisoning, the attacker has to run the training model much more times than the baseline algorithms, making the proposed algorithm less practical. However, the paper touches little about this topic, and does not provide any comparison in the experiment section. I think this information is crucial for the readers to better understand and appreciate the proposed algorithm.

2. Multiple challenge points. Usually in practice, the attacker needs to attack multiple challenge points instead of the only one. Although the paper briefly discusses this in the appendix, I think it is far from enough. Specifically, Algorithm 2 is just a simple generalization of Algorithm 1, neglecting many interesting and important problems due to more than one challenge points. For example, the problem of time complexity becomes even worse. Furthermore, due to the correlations of different challenge points, it is not clear how Algorithm 2 performs. Considering an extreme case when there are two challenge points opposing each other, it is possible after k_max iterations, the algorithm can not find meaningful k_i for both points simultaneously.

3. Clarity (minor points). The paper needs to improve the clarity. For example, many definitions are used without being defined, e.g., LIRA, challenge point, in+out model. It is better to provide those definitions in the preliminary to make the paper more self-contained.

**Questions:**

Please refer to the section above.

---

> ### Author Response · Authors · 2023-11-15
>
> We thank the reviewer for their comments and suggestions.  We resonate strongly with the comment on the importance of label-only membership inference in private ML, and we hope that our paper provides a step forward in this area.  We address the main concerns below.  In the paper, we marked the modified text in blue, for ease of identification.
>
> **Time Complexity:**
>
> While we acknowledge that shadow model training incurs a cost, it is crucial to note that our attack trains a fixed number of shadow models, independent of  the number of challenge points. In our experiments for CIFAR-10, inferring membership over 500 challenge points simultaneously, our attack trains a total of 80 shadow models and achieves a 20.8x TPR(@1%FPR) improvement over the Decision-Boundary (DB) attack. Despite the training cost, it still remains notably lower than that of state-of-the-art confidence based membership inference attacks like LiRA (Carlini et al., 2022) and TruthSerum (Tramer et al., 2022), which typically require 128 shadow models for optimal performance.
>
> Additionally, by careful selection of hyperparameters in Algorithm 2, our attack achieves a 12.1x TPR(@1%FPR) improvement over the DB attack while requiring to train only 16 shadow models. In Appendix D, we provide a detailed cost analysis associated with running our attack along with end-to-end running time comparison between ours and DB attack. Interestingly, our computationally constrained variant, using 16 models, nearly matches the runtime of the DB attack while significantly outperforming it in terms of TPR (12.1x improvement).
>
> **Multiple Challenge Points:**
>
> We would like to clarify that Algorithm 2 (Appendix C) does address the drawbacks highlighted by the reviewer.  Algorithm 2 indeed handles multiple challenge points simultaneously while training only a fixed number of shadow models, independent of the number of challenge points, and also captures the potential interactions between different challenge points and their respective poisoned replicas. We have updated Appendix C to explicitly elaborate on the impracticality of a straightforward extension of Algorithm 1 (also pointed out by the reviewer) and how Algorithm 2 can be used as an efficient alternative to tackle these drawbacks.
>
> **Clarity:**
>
> Thank you for highlighting this issue. To ensure the paper is as self-contained as possible, we have added definitions of the terms used throughout the paper in sections 2 and 3.
>
> We hope our response has addressed the main concerns expressed in the review. We will be happy to provide additional clarifications if needed.

---

### Meta-Review · Area_Chair_ZUuV · 2023-12-11

**Metareview:**

This paper introduces a data poisoning attack called Chameleon. Unlike prior membership inference attacks that rely on a model's predicted confidence scores, Chameleon operates in a label-only setting, where attackers have access only to the predicted label of a queried sample. Surprisingly, label-only MI attacks prove ineffective when only labels are available, especially at low FPRs. Chameleon overcomes this by employing a novel adaptive poisoning strategy to enhance membership inference leakage in the label-only setting combined with an efficient querying strategy that uses significantly fewer queries compared to competing methods (dozens instead of hundreds).

Overall, this is an interesting paper that raises the bar for membership inference attacks. My own reading of the paper confirms the favourable reviews. It's a bit unfortunate that the reviewers did not engage in the discussion with the authors.

I do note that the proposed method (Chameleon) has limitations raised in the reviews that could be more clearly acknowledged in the paper. These include improving the clarity of the theoretical results and acknowledging the complexity of the proposed attack.

**Justification For Why Not Higher Score:**

Though compelling, the proposed Chameleon attack has limitations noted by the reviewers (e.g., time complexity). Moreover, as also noted by the reviewers, the clarity of the paper can be improved.

**Justification For Why Not Lower Score:**

This is a solid contribution to the broader field or membership inference attacks.

---

### Decision · Program_Chairs · 2024-01-16

Accept (poster)